# Physical activity interventions for the cardiac population: A generic logic model based on intervention mapping

**Tamika Akesha Marcos**[1,2]✉*, **Stefan Tino Kulnik**[2]✉, **Rik Crutzen**[2,3]✉

**1** Department of Work and Social Psychology, Maastricht University, Maastricht, The Netherlands, **2** Ludwig Boltzmann Institute for Digital Health and Prevention, Salzburg, Austria, **3** Department of Health Promotion, Care and Public Health Research Institute, Maastricht University, Maastricht, The Netherlands

✉ These authors contributed equally to this work.
* t.marcos@maastrichtuniversity.nl

## Abstract

Regular physical activity constitutes a crucial secondary prevention behaviour for people who have had an acute cardiac event. Nonetheless, sustained physical activity levels of the cardiac population are in need of improvement. Applying the Intervention Mapping approach, we describe a generic logic model for developing behavioural interventions to increase and maintain physical activity among those who have had an acute cardiac event. The development of the logic model was both data- and theory-driven, which enhances the effectiveness of the behavioural interventions developed using the logic model. The logic model may additionally be used to indicate research gaps, to solidify the underpinnings of already established behavioural interventions, and as a reflection tool for healthcare providers and individuals' behaviour. A generic logic model like the one presented here can be developed for key behavioural outcomes related to preventing other highly relevant health problems. We further recommend that future empirical research focuses on determinants of social support provision for cardiac patients' physical activity.

## Introduction

Cardiovascular diseases (CVDs) are the leading cause of death worldwide, with 8.9 million deaths attributed to coronary artery disease and 180.9 million healthy life years lost to CVDs in 2019 [1]. After an acute cardiac event, exercise-based cardiac rehabilitation offers an important care pathway for secondary prevention, reducing the risk of further fatal and non-fatal cardiac events and increasing quality of life [2,3].

Exercise-based cardiac rehabilitation is a supervised, multi-faceted programme delivered by healthcare providers to address behavioural risk factors for CVD, such as unhealthy diet, tobacco use, and harmful use of alcohol, with a focus on physical inactivity. The aim is to support patients in establishing a habit of performing

**Data availability statement:** No empirical data was collected for this research. This research is based on evidence from previously published research. These articles are publically available in their respective journals

**Funding:** The author(s) received no specific funding for this work.

**Competing interests:** The authors have declared that no competing interests exist.

regular (weekly) physical activity at recommended heart-healthy levels [3]. As a general recommendation, patients should conduct endurance training at moderate to vigorous intensity at least three times per week and ideally six to seven times per week. In addition, muscle-strengthening exercises for the large muscle groups should be performed twice weekly. Physical activity should correspond to a caloric consumption of 1,000–2,000 kcal per week [3]. For many patients with CVD who previously led an inactive lifestyle, this means increasing the weekly amount of physical activity.

Physical activity is defined as any bodily movement produced by skeletal muscles that results in energy expenditure at work or during leisure time, such as sports, conditioning exercises, household tasks, and other activities [4]. The term physical activity is used here as the overarching descriptor for different sub-types of physical activity, such as exercise (i.e., physical activity that is planned, structured, repetitive and purposive for improving or maintaining physical fitness) [4].

Notably, following the completion of a cardiac rehabilitation programme, patients should adhere to a heart-healthy lifestyle for the remainder of their lives. They should independently maintain regular physical activity at the recommended heart-healthy levels, e.g., by continuing with independent endurance training at home or by joining exercise classes for CVD patients or other suitable offers. However, this long-term habit formation and continuation of regular heart-healthy physical activity often fails, as demonstrated in the 2017 EUROASPIRE V survey of 8,261 cardiac patients in 27 European countries [5]. On average (median) 1.1 years (interquartile range 0.8–1.6 years) following an acute coronary event, only 34% of patients reported performing regular physical activity (i.e., ≥30 minutes on average five times a week); only 16% reported performing high-intensity physical activity at least three times a week for ≥20 minutes; only 35% reported performing planned physical activity to increase physical fitness; 42% of patients did not perform regular physical activity and had no intention to do so in the next six months; and 46% of patients did not recall having received personal advice on physical activity [5].

A systematic review published in 2015 examined the impact of attending a cardiac rehabilitation programme on patients' medium-term (up to six months) and long-term (over six months) heart-healthy physical activity behaviour after completion of the programme [6]. The review included 26 randomised controlled trials. The organisation of the cardiac rehabilitation programmes examined in these studies was heterogeneous in terms of duration (4 weeks to 4 years) and format (centre-based versus home-based). The review concluded that having completed a cardiac rehabilitation programme did not show a clear positive impact on sustained heart-healthy physical activity behaviour compared with patients who did not participate in such a programme. The duration of the rehabilitation programme or the volume of exercise in the rehabilitation programme did not show any association with sustained heart-healthy physical activity behaviour. However, home-based rehabilitation programmes appeared to be slightly better than centre-based programmes at contributing to heart-healthy physical activity habit formation [6].

In summary, the evidence describes the urgent and challenging problem of sustained behaviour change (long-term habit formation) for independent regular heart-healthy physical activity, exercise, training, or sports in patients who have suffered an acute cardiac event.

## Aim

Given the importance of maintaining regular physical activity, especially after a cardiac event, numerous efforts exist to increase the maintenance of physical activity for cardiac patients [7]. This is a difficult behavioural challenge and requires addressing a range of factors on multiple levels [7,8]. Hence, we need to gain further insights on which aspects to tackle and how for future intervention efforts. Therefore, this research aimed to create a generic logic model, which can serve as a blueprint for developing interventions that increase (the maintenance of) physical activity among patients who have had an acute cardiac event using the Intervention Mapping approach [9]. Logic models provide a transparent map of an intervention, explicitly uncovering its underlying assumptions and its proposed or hypothesized mechanisms of action. Methodological guidance for the development and evaluation of behavioural interventions (Template for Intervention Description and Replication (TIDieR) checklist [10], United Kingdom Medical Research Council's framework for developing and evaluating complex interventions [11], Workgroup for Intervention Development and Evaluation Research (WIDER) [12]) recommends the use of logic models. Therefore, developers and investigators of behaviour change interventions are urged to – and often do – report logic models (e.g., [13]). Despite that, these are usually specific to the intervention, they intend to convey a comprehensive understanding of that specific intervention. A generic logic model as proposed in this research, will facilitate and catalyse the development of effective theory- and evidence-based interventions. In addition, it helps to identify research gaps where insights still need to be improved. We furthermore anticipate that this generic logic model will be a valuable resource for researchers and intervention developers constrained by limited time and resources, enabling them to engage with the Intervention Mapping approach.

## The intervention mapping approach

Intervention Mapping (IM) is an established approach that supports health promotion programme planners in developing, implementing, and evaluating health promotion interventions. The IM approach starts from available theory and evidence, taking an ecological approach to addressing health issues and promoting community involvement [9].

IM consists of six steps that entail various tasks. Upon completion, each step provides a guide for the following step [9]. The first step of IM creates a logic model of the problem based on the PRECEDE model by Green and colleagues [14]. This logic model of the problem is developed for both an individual (patient) behavioural perspective and an environmental perspective. In step two, the focus shifts from a problem-orientated to a solution-orientated view, and a logic model of change is developed. The logic model of change describes what needs to change in the behaviour at the individual and environmental levels.

The behaviour change at the environmental levels is to be accomplished by so-called environmental agents (e.g., healthcare providers, but also spouses and friends). Through their behaviour, these environmental agents have the power and ability to exert influence on the environment and, therewith, on the behaviour of the individual patient. Step two is completed by formulating performance and change objectives (POs and COs, respectively). POs can be interpreted as sub-behaviours and have determinants which explain the behaviour. Combining the POs with their corresponding determinants will result in the formulation of COs. COs serve as targets for practical applications in the intervention. These practical applications are based on behaviour change methods selected in the third step of IM. The fourth step entails the development of the intervention structure and organisation and designing, developing, and pre-testing the materials and programme. Step five comprises of developing adoption and implementation plans, and the final step, step six, completes the evaluation plan of the developed behaviour change intervention. Completing all steps establishes a context-specific intervention grounded in theoretical, empirical and practical knowledge and data [9].

## Development of the generic logic model

The development of the generic logic model was guided by the first two steps of IM [9]. The at-risk population in this logic model are patients who have had an acute cardiac event. Healthcare providers (e.g., doctors, nurses, sports scientists, physiotherapists) [15] and people in the patient's interpersonal environment (e.g., spouse, other family members, friends) [16,17] were chosen as two key environmental agents, due to their influence on the behaviour and (social) environment of the patients.

## The logic model of the problem

The logic model of the problem (Fig 1) was established following the PRECEDE model, as reported by Green et al. [14]. Insufficient independent long-term maintenance of heart-healthy physical activity was taken as the at-risk behaviour for the individual. Conraads et al.[18] reported inadequate social support as an external barrier to participation in physical activity among the cardiac population. Therefore, a lack of support provision for long-term independent physical activity from the healthcare providers and people in the patient's interpersonal environment were identified as the environmental factors for the cardiovascular-related health problems and negative quality of life effects.

On the individual level, fear that being physically active will aggravate heart issues, or kinesiophobia, was found to have a strong to very strong (r ranges from −0.56 to −0.80) association with less physical activity behaviour in several studies [19–21]. Le Grande and colleagues [22] also found being persistently depressed to be strongly associated with less physical activity. Peersen et al. [23], Ali et al. [24], Blanchard et al. [25], Kronish et al. [26] and van der Wal et al. [27] all

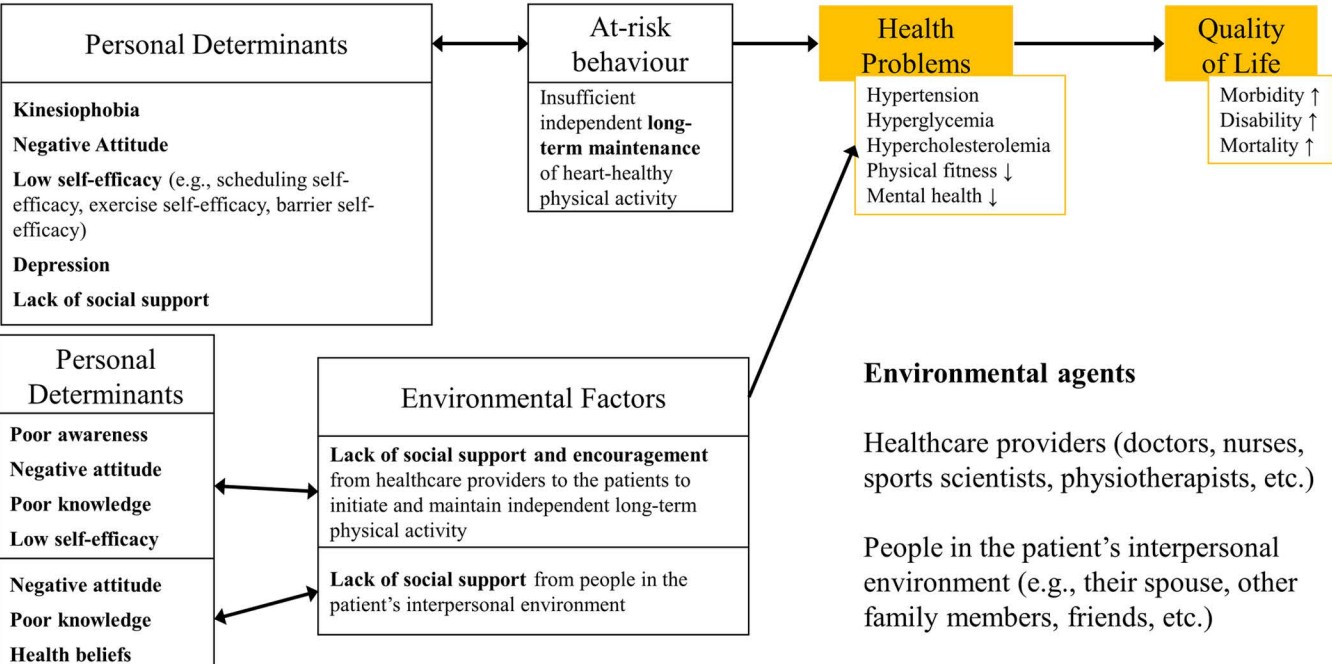

**Fig 1. Logic model of the problem.**

reported a weak to moderate association between depression and less physical activity. Self-efficacy is a well-established correlate of behaviour according to several behaviour theories. Ali et al. [24], Blanchard et al. [25], De Melo Ghisi et al. [28], Klompstra et al. [29], Luszczynska and Sutton [30], Murray and Rodgers [31], and Slovenic D'Angelo et al. [32] all found lower self-efficacy to lead to less physical activity and higher self-efficacy to more physical activity among the cardiac population. In a Bayesian meta-analysis on physical activity and heart failure, Amirova and colleagues [33] reported that a negative attitude towards physical activity is a barrier to physical activity among those with heart failure.

A systematic review conducted by Daw and colleagues [8] reported poor knowledge of, poor awareness of and negative attitude towards cardiac rehabilitation programmes as a barrier for healthcare providers to refer their patients and thus support and encourage them to be physically active. This is further supported by the general behaviour change pathway developed by Williamson and colleagues [34]. Birtwistle and colleagues [17] stated that beliefs of families that physical activity might negatively affect the patient's heart health obstructs the patient's physical activity.

## The logic model of change

Based on the logic model of the problem, the individual and environmental outcomes were re-formulated by changing the focus from problem-orientated to solution-orientated (Fig 2) [9]. The individual outcome for patients is to regularly carry out the recommended amount of heart-healthy physical activity and maintain this behaviour in the long term. The environmental outcomes for both environmental agents are to support and encourage the patient to be physically active.

The core processes embedded in IM were used to formulate the POs and COs and uncover the determinants. There are six core processes: 1) posing questions, 2) brainstorming, 3) reviewing findings from the empirical literature, 4) reviewing theories for additional constructs, 5) assessing and addressing needs for new data, and 6) developing a working list of answers [9]. These core processes need to be conducted in this order, and each must be completed before one continues to the next question. The core processes ensure the problem is completely understood, enhancing the solution's effectiveness [35].

The POs were determined by asking the questions: "What do people who had an acute cardiac event need to do to increase and maintain their heart-healthy physical activity?" and "What do the healthcare providers and people in the patient's interpersonal environment need to do to encourage and support the patient in being physically active?". The POs specify what actions are required of the participants in the intervention. Determinants were correspondingly selected based on the questions: "Why would people increase and maintain their heart-healthy physical activity?" and "Why would healthcare providers and people in the patient's interpersonal environment encourage and support the patients in being physically active?". Determinants were then rated based on their importance (the strength of association with the behaviour) and changeability (the likelihood of interventions being able to influence the determinant) [9]. This process was informed by empirical evidence from a systematic review of determinants of physical activity in cardiac patients [36,37]

After posing the questions, a brainstorm using free association and the researchers' expertise yielded an initial list of potential answers. For example, the question "What do people who had an acute cardiac event need to do to increase and maintain their heart-healthy physical activity?" was answered by "Adjusting exercises to their physical capabilities". The subsequent core process in which existing research and literature were reviewed, validated or disregarded these answers. In the next core process, to provide a more complete answer to the questions, general behaviour theories such as, but not limited to, The Reasoned Action Approach [38], the Health Action Process Approach [39], the I-Change Model [40,41] and the Social-Ecological model [42] were consulted. Research gaps were identified hereafter, and a final list of answers was established. These processes were then repeated to formulate and uncover the other POs, COs, and determinants.

After the final selection of determinants, the POs and their linked determinants were combined to construct the matrices of COs. Table 1 shows examples of POs for the individual behavioural outcome and their linked determinants and COs.

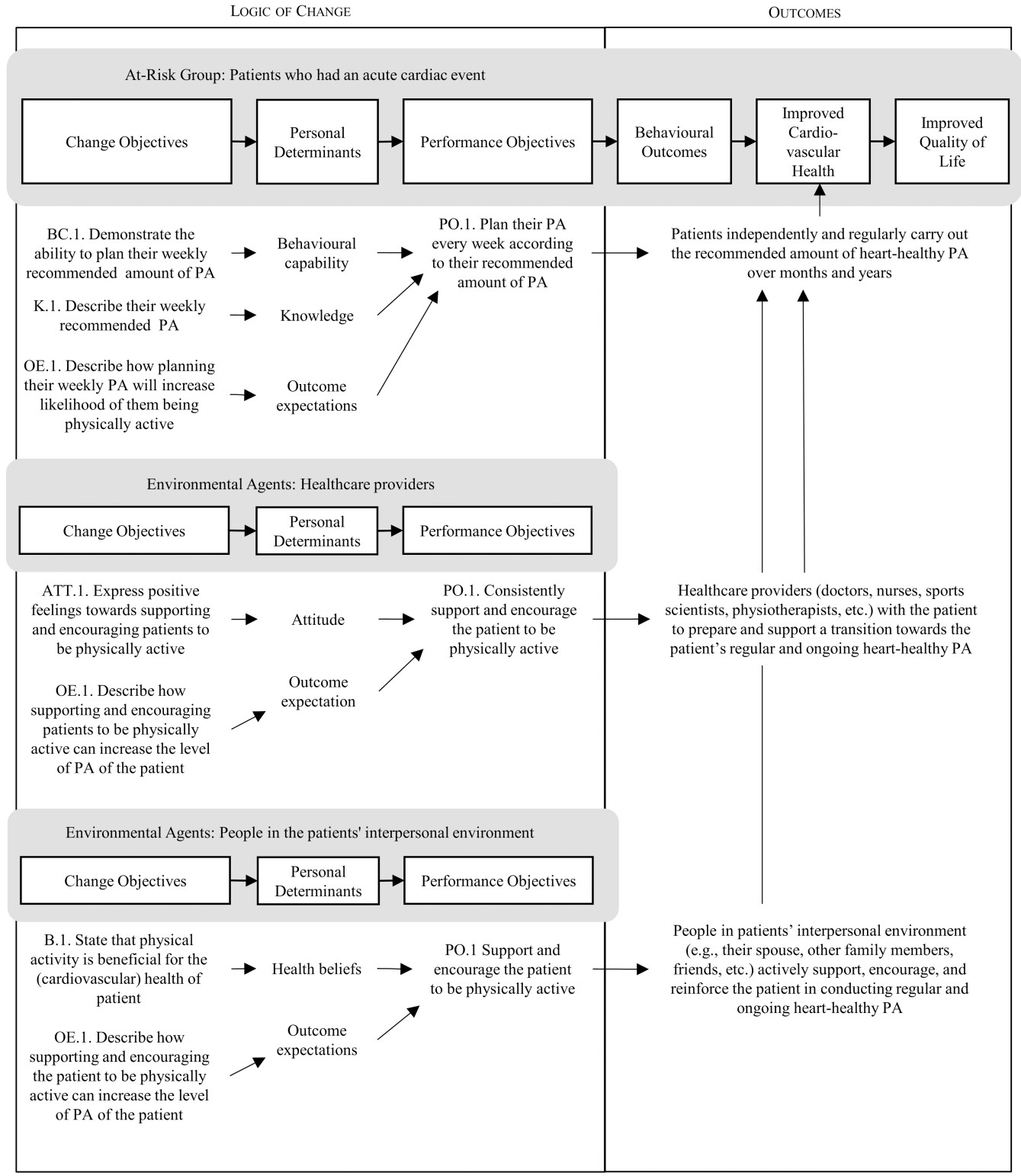

**Fig 2. Logic model of change.**

**Table 1. Examples of COs for selected POs for the individual behavioural outcome.**

| Performance objective | Determinant | Change objective |
|---|---|---|
| PO.1. Plan their physical activity every week according to their recommended physical activity [43] | Behavioural capability Knowledge Outcome expectations | BC.1. Demonstrate the ability to plan their weekly recommended physical activity [44] <br> K.1. Describe their weekly recommended physical activity [38] <br> OE.1. Describe how planning their weekly physical activity will increase likelihood of them being physically active [43] |
| PO.4. Self-monitor their physical activity using a log or diary [6,7,45–48] | Attitude Outcome expectations Skills | ATT.4. Express positive feelings towards self-monitoring physical activity [38] <br> OE.4. Describe how monitoring of physical activity will increase the likelihood of maintaining regular physical activity [49] <br> S.4. Demonstrate the ability to monitor own physical activity using a log or diary [38] |

## Results

The detailed generic logic model is described in an additional Microsoft Excel file [see S1 File]. The S1 File contains three worksheets relating to the individual patient and three relating to the environmental agents (healthcare providers and people in the patient's interpersonal environment). The worksheets labelled "ABCD" list the target population(s), the target behaviour(s), the POs, determinants, the COs, behavioural change methods, parameters for use, and examples of practical implications. The 'ABCD' refers to the acyclic behaviour change diagram, a tool to promote clarity and transparency in developing, evaluating, and reporting interventions [50]. The works labelled "Selection of determinants" list the decisions to select, defer or forgo the found determinants based on their importance and changeability, and the worksheets labelled "CO matrix" list all COs categorised per PO and determinant.

Altogether, the generic logic model includes seven POs, eleven determinants, and twenty-nine COs for the individual behavioural outcome and eleven POs, six determinants, and twenty-three COs for the environmental behavioural outcomes. Altogether, forty scientific publications support the model. An overview of all POs and determinants can be found in Table 2.

In addition to the Microsoft Excel file [S1 File], an interactive clickable version of the generic logic model is provided in S2 File. This offers a more intuitive presentation of the logic model and additionally includes examples of methods and practical applications to address individual COs.

## Discussion

Using the IM approach, this study developed a generic logic model by summarising known determinants of physical activity initiation and maintenance among the cardiac population. To our knowledge, this is the first generic application of an IM logic model, offering a novel approach to employing IM methodology. The model provides a blueprint for developing interventions aiming to increase and maintain the physical activity of cardiac patients. Utilising the generic logic model aligns with the United Kingdom Medical Research Council (MRC) guidance, which emphasises the importance of supporting the development of behaviour change interventions through evidence- and theory-based logic models [11].

### Practical applications and recommendations

The generic blueprint serves as a collection of current evidence regarding increasing and maintaining physical activity among the cardiac population and can be used in various situations. Next, we will provide some examples and discuss how the blueprint can facilitate the desired behaviour change among the cardiac population, their healthcare providers, and their interpersonal social circles.

**Developing new interventions.** A generic blueprint provides a robust foundation for developing theory- and evidence-based interventions, enabling intervention developers to build upon established logic models and the available insights in the current literature. This approach not only catalyses the intervention development process but also reduces the demand for resources, ultimately enhancing the accessibility of theory- and evidence-based interventions. An intervention

**Table 2. Target populations, POs, and determinants.**

| Target populations | Performance objectives | Determinants |
|---|---|---|
| Patients who had an acute cardiac event | Plan their physical activity every week according to their recommended amount of physical activity<br>Carry out their heart-healthy physical activity according to their weekly plan<br>Manage potential barriers for planned physical activity sessions [e.g., physical capabilities, weather conditions, lack of time]<br>Self-monitor their physical activity using an activity log or a diary<br>Adjust their physical activity plan when their health status improves or diminishes<br>Adjust their physical activity plan when their exercise capacity improves or diminishes<br>Seek professional and/or medical help when unsure what exercises to do/how to adjust exercises | Attitude<br>Behavioural capacity<br>Kinesiophobia<br>Knowledge<br>Motivation<br>Outcome expectations<br>Perceived barriers and benefits<br>Self-efficacy<br>Social influence<br>Subjective needs |
| Healthcare providers (doctors, nurses, sports scientists, physiotherapists, etc.) | Consistently support and encourage the patient to be physically active<br>Provide performance feedback on the physical activity of the patient<br>Assist the patient in planning their physical activity<br>Refer the patient to a cardiac rehabilitation programme regardless of their age, sex, previous physical activity levels, and socioeconomic position<br>Educate the patient on the health benefits of regular physical exercise<br>Encourage the patient to resume physical activity after a lapse<br>Assess the current and prior (before the acute episode of care) physical activity level of the patient | Attitude<br>Awareness<br>Knowledge<br>Outcome expectations<br>Perceived barriers<br>Skills |
| People in the patients' interpersonal environments (e.g., their spouses, other family members, friends etc.) | Support and encourage the patient to be physically active<br>Join the patient in their weekly planned physical activity<br>Assist the patient in planning physical activity adjusted to their needs<br>Support the patient in managing potential barriers | Attitude<br>Health beliefs<br>Knowledge<br>Outcome expectations |

developer uses the generic blueprint by selecting POs and COs relevant to their particular context and completes the logic model by determining and choosing appropriate methods and practical applications for their intervention. Health promotion has an everlasting fidelity vs adaptation debate in the field of health promotion [51,52]. Adaptation of intervention content is unavoidable when transferring to a different setting. However, ensuring the integrity and effectiveness of an intervention remains crucial. The balance between fidelity and adaptation, or 'navigating the middle' as Paton [53] describes it, can take much work. Therefore, collaboration with a diverse planning group in selecting context-relevant POs and COs is necessary. To ensure a complete and effective intervention, developers are advised to continue planning their intervention according to the IM steps [9]. Stutterheim and colleagues [54] provide an exemplary elaboration on what completing these steps looks like in practice.

**Solidifying existing interventions.** The generic blueprint can be beneficial in solidifying the theory- and evidence-based foundations for already existing interventions. For example, an intervention developed without an explicit underpinning logic model may be mapped against the blueprint to 'retrofit' an intervention-specific logic model. This was demonstrated by Marcos et al. [55], who used the generic logic model developed in this study, to map the underlying rationale behind the aktivplan programme. Equally, an existing intervention and its underpinning logic model may be mapped and compared against the generic blueprint to identify the potential for improvement or additions with the aim of increasing intervention effectiveness. Bartholomew Eldredge et al. [9] additionally describe how the IM approach can adapt established evidence-based interventions to new contexts, which can guide solidifying and adapting established interventions.

**Identifying research gaps.** For researchers, the generic blueprint facilitates the identification of research gaps as it provides a clear and concise overview of the knowledge and evidence currently available on the problem. The evidence for selecting the determinants provides insight into the determinants lacking empirical evidence and thus identifies areas where more research is needed. An example derived from the generic logic model is the determinants of social support provision for those in the interpersonal environment of the patient. No empirical research regarding the psychological determinants of support provision was found.

**Reflecting on one's behaviour.**  Healthcare providers can use the generic logic model as a reflection tool for their interactions with cardiac patients. Healthcare providers can compare their current behaviour to the desired behaviour stated in the generic blueprint and adjust accordingly. Patients and those in their interpersonal environments can similarly use the generic logic model (independently or guided by a healthcare professional or health coach) to reflect on and, where necessary, adjust their behaviour. Readers are highly encouraged to augment the additional Microsoft Excel file (S1 File) suitable to their specific behavioural and environmental contexts.

## Limitations

The development of this generic logic model was guided by the established IM approach with two noteworthy limitations. The logic model was not developed in collaboration with a formal planning group, which is usually recommended in particular for the brainstorming process (as part of the core processes). However, establishing a formal planning group would not align with the generality of this logic model, as a planning group mainly depends on the specific context in which an intervention will be applied. This generic blueprint provides a complete and relevant overview of the essential and related determinants, POs, and COs. We strongly recommend that when this logic model is applied in a specific context, a formal planning group consisting of members who are relevant to the setting is involved to ensure appropriate contextualisation and high-quality outcomes.

A further limitation of this generic logic model is that while the outcomes, POs, and COs were focused on two vital environmental agents at the interpersonal level, other environmental agents at the remaining levels of the social-ecological model might also be necessary. Given their inherent variability across settings, the organisational, policy and societal levels were disregarded. To develop complete interventions, intervention developers should incorporate these social-ecological levels accounting for their specific environmental context.

## Future directions

The evidence for some of the determinants presented in the logic model was based on generic behaviour theories. This often occurred for determinants lacking empirical research, particularly environmental agents. Whilst the unequal amount of research on determinants of physical activity-related behaviour for the cardiac patient versus the environmental agents is understandable and justifiable, we encourage future research to include research on the behavioural determinants of those in the broader social environments. For example, literature on the determinants of social support provision was extremely scarce. Hence, there is a necessity for future research to focus on what exactly motivates people to provide support to those in their social and professional environments.

In this article, we developed a generic logic model using the IM approach to increase and maintain physical activity levels among those with an acute cardiac event. The development of such a generic logic model can be applied to key behavioural outcomes related to the prevention of other health problems that affect a variety of people as well. By developing such a generic logic model, researchers can catalyse the development of effective and theory- and evidence-based interventions to improve the health and quality of lives of the diverse populations in their respective fields.

## Supporting Information

**S1 file.**
(XLSX)

**S2 file.**
(PPTX)

## Acknowledgments

Large Language Model ChatGPT has been consulted to inspire practical examples for the additional materials.

## Author contributions

**Conceptualization:** Tamika Akesha Marcos, Stefan Tino Kulnik, Rik Crutzen.

**Investigation:** Tamika Akesha Marcos, Stefan Tino Kulnik.

**Writing – original draft:** Tamika Akesha Marcos.

**Writing – review & editing:** Tamika Akesha Marcos, Stefan Tino Kulnik, Rik Crutzen.

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
