## [Decision Letter · Decision Letter 0]

14 May 2024

PONE-D-23-43225A Generic Logic Model for the Development of Physical Activity Interventions for the Cardiac Population Using Intervention MappingPLOS ONE

Dear Dr. Marcos,

Thank you for submitting your manuscript to PLOS ONE. After careful consideration, we feel that it has merit but does not fully meet PLOS ONE’s publication criteria as it currently stands. Therefore, we invite you to submit a revised version of the manuscript that addresses the points raised during the review process.

We look forward to receiving your revised manuscript.

Kind regards,

Tamer I. Abo Elyazed, Ph.d

Guest Editor

PLOS ONE

Journal Requirements:

Additional Editor Comments:

Your manuscript title should include the type of the study.

Reviewers' comments:

Reviewer's Responses to Questions

**Comments to the Author**

1. Is the manuscript technically sound, and do the data support the conclusions?

Reviewer #1: Yes

Reviewer #2: Partly

2. Has the statistical analysis been performed appropriately and rigorously? 

Reviewer #1: I Don't Know

Reviewer #2: N/A

3. Have the authors made all data underlying the findings in their manuscript fully available?

Reviewer #1: Yes

Reviewer #2: Yes

4. Is the manuscript presented in an intelligible fashion and written in standard English?

Reviewer #1: Yes

Reviewer #2: Yes

5. Review Comments to the Author

Reviewer #1: the generic model must be translated into different languages

the authors did not mention the number of cases they applied the model on

the model must be of reliability and validity tests.

the references must be in the same form .....doi ..................

Reviewer #2: Please add few examples how logic models have previously been used and successful in sustainability of the behavior

Need more clarification on how the core processes embedded in IM were used to formulate the POs and COs and

uncover the determinants.

Results need further elaboration with practical examples

6. PLOS authors have the option to publish the peer review history of their article (what does this mean? ). If published, this will include your full peer review and any attached files.

**Do you want your identity to be public for this peer review?** For information about this choice, including consent withdrawal, please see our Privacy Policy .

Reviewer #1: No

Reviewer #2: No

---

## [Author Response · Author response to Decision Letter 0]

27 Jun 2024

Dear Editor and reviewers,

Thank you for the opportunity to revise and resubmit our manuscript. We have carefully considered the reviewers' comments and incorporated the necessary changes in the revised manuscript. A detailed response to each reviewer's comments can be found in the uploaded file titled "Response to Reviewers." This document outlines the specific revisions made and addresses any raised concerns.

Thank you again for your time and consideration.

Sincerely,

The authors.

---

## [Decision Letter · Decision Letter 1]

7 Oct 2024

PONE-D-23-43225R1Physical activity interventions for the cardiac population: a generic logic model based on Intervention MappingPLOS ONE

Dear Dr. Marcos,

Thank you for submitting your manuscript to PLOS ONE. After careful consideration, we feel that it has merit but does not fully meet PLOS ONE’s publication criteria as it currently stands. Therefore, we invite you to submit a revised version of the manuscript that addresses the points raised during the review process. Please submit your revised manuscript by Nov 21 2024 11:59PM. If you will need more time than this to complete your revisions, please reply to this message or contact the journal office at plosone@plos.org . Please include the following items when submitting your revised manuscript:

We look forward to receiving your revised manuscript.

Kind regards,

Tamer I. Abo Elyazed, Ph.d

Guest Editor

PLOS ONE

Journal Requirements:

Reviewers' comments:

Reviewer's Responses to Questions

**Comments to the Author**

1. If the authors have adequately addressed your comments raised in a previous round of review and you feel that this manuscript is now acceptable for publication, you may indicate that here to bypass the “Comments to the Author” section, enter your conflict of interest statement in the “Confidential to Editor” section, and submit your "Accept" recommendation.

Reviewer #2: All comments have been addressed

Reviewer #3: (No Response)

2. Is the manuscript technically sound, and do the data support the conclusions?

Reviewer #2: Yes

Reviewer #3: Yes

3. Has the statistical analysis been performed appropriately and rigorously? 

Reviewer #2: N/A

Reviewer #3: N/A

4. Have the authors made all data underlying the findings in their manuscript fully available?

Reviewer #2: Yes

Reviewer #3: Yes

5. Is the manuscript presented in an intelligible fashion and written in standard English?

Reviewer #2: Yes

Reviewer #3: Yes

6. Review Comments to the Author

Reviewer #2: (No Response)

Reviewer #3: This is a very complex piece of work, and (I appreciate) difficult to describe within the remit of a journal article. It explains a process which is rarely described in the literature, and I have appreciated the opportunity to read it. The initial reviewer comments have been addressed. Referring to the revised document, I have some minor comments:

Line 93 'a range of factors' would be a better phrase than 'various factors' (which makes this sentence feel vague).

Line 128 Could you say 'the individual patient' rather than 'the individual' - to be specific (as the patient is another 'agent')?

Lines 141-5 I had to re-read this paragraph more than once to understand it. You finish the first sentence by specifying a population, and begin the following sentence with a reference to 'their influence' (the patients?). If you put the subject of the following sentence first - 'People in the patient's interpersonal environment...because of their influence on the patient's behaviour and social environment' - this will read grammatically.

My main comment is that I was expecting to see a figure showing the logic model of change (the solution-oriented model you would apply in intervention development) rather than (or as well as) the logic model of the problem. This appears in the Marcos et al paper you've now cited. Table two does outline the constructs associated with change, but it would be very helpful to the reader to have a concise diagrammatic representation of the change model (many readers might not open your supplementary files).

7. PLOS authors have the option to publish the peer review history of their article (what does this mean? ). If published, this will include your full peer review and any attached files.

**Do you want your identity to be public for this peer review?** For information about this choice, including consent withdrawal, please see our Privacy Policy .

Reviewer #2: No

Reviewer #3: No

---

## [Author Response · Author response to Decision Letter 1]

18 Nov 2024

Please find our detailed, point-by-point responses to the reviewers’ comments included in the document titled 'Response to reviewers'

---

## [Editor Report · Decision Letter 2]

28 Mar 2025

Physical activity interventions for the cardiac population: a generic logic model based on Intervention Mapping

PONE-D-23-43225R2

Dear Dr. Marcos,

We’re pleased to inform you that your manuscript has been judged scientifically suitable for publication and will be formally accepted for publication once it meets all outstanding technical requirements.

Kind regards,

Jianhong Zhou

Staff Editor

PLOS ONE
---

## [Editor Report · Acceptance letter]

PONE-D-23-43225R2

PLOS ONE

Dear Dr. Marcos,

I'm pleased to inform you that your manuscript has been deemed suitable for publication in PLOS ONE. Congratulations! Your manuscript is now being handed over to our production team.

Kind regards,

on behalf of

Dr. Jianhong Zhou

Staff Editor

PLOS ONE